# Angular difference in human coronary artery governs endothelial cell structure and function

Yash T. Katakia[1], Satyadevan Kanduri[2], Ritobrata Bhattacharyya[1], Srinandini Ramanathan[1], Ishan Nigam[1], Bhanu Vardhan Reddy Kuncharam [2✉] & Syamantak Majumder [1✉]

Blood vessel branch points exhibiting oscillatory/turbulent flow and lower wall shear stress (WSS) are the primary sites of atherosclerosis development. Vascular endothelial functions are essentially dependent on these tangible biomechanical forces including WSS. Herein, we explored the influence of blood vessel bifurcation angles on hemodynamic alterations and associated changes in endothelial function. We generated computer-aided design of a branched human coronary artery followed by 3D printing such designs with different bifurcation angles. Through computational fluid dynamics analysis, we observed that a larger branching angle generated more complex turbulent/oscillatory hemodynamics to impart minimum WSS at branching points. Through the detection of biochemical markers, we recorded significant alteration in eNOS, ICAM1, and monocyte attachment in EC grown in microchannel having 60º vessel branching angle which correlated with the lower WSS. The present study highlights the importance of blood vessel branching angle as one of the crucial determining factors in governing atherogenic-endothelial dysfunction.

[1] Department of Biological Sciences, Birla Institute of Technology and Science (BITS), Pilani Campus, Pilani, India. [2] Department of Chemical Engineering, Birla Institute of Technology and Science (BITS), Pilani Campus, Pilani, India. ✉email: bhanu.vardhan@pilani.bits-pilani.ac.in; syamantak.majumder@pilani.bits-pilani.ac.in

A therosclerosis is the underlying cause of more than 50% of cardiovascular afflictions and is thus a major contributor to deaths. Globally, ischaemic heart disease and strokes caused due to atherosclerosis take an estimated 17.9 million lives each year (32% of all deaths)[1]. The initiation and development of atherosclerosis are primarily determined by endothelial well-being. The endothelium lining the cardiovascular system is highly sensitive to hemodynamic forces that act at the vessel luminal surface in the direction of blood flow. While several mechanical and biochemical factors regulate endothelial cells (EC), one of the most important regulators of endothelial functions as wall shear stress (WSS)[2]. In arteries, physiological levels of shear stress range from 10–70 dyn/cm$^2$, and steady laminar flow (S-Flow) is associated with increased expression of genes beneficial to EC homeostasis[3]. In contrast, in areas of vessel curvature and branching, shear stress is significantly lower (<4 dyn/cm$^2$), and the flow pattern is disturbed (D-Flow), characterized by recirculation and oscillations[4,5]. Transient, unstable flow separation that creates flow disturbance regions containing oscillating, transient vortices is associated with a predisposition to atherosclerosis at branches, bifurcations and curvatures in the arterial circulation.

With such pivotal roles, studies have accentuated the impact of vessel geometry, velocity distribution and the associated WSS on the localization, progression and clinical outcomes of atheroma development. In a pioneering effort (1969), Caro et al. drew a correlation between arterial flow mechanics and atherosclerotic plaque formation[6]. Successive efforts concluded that the dispensation of early atheroma in humans aligns with the vessel branch points, which experience substantially reduced wall shear rates[7–10]. Analyses through computational fluid dynamics (CFD) simulations have enabled highly precise and clinically relevant ascertainment of such hemodynamic influencers. Numerical simulations with concurrent experimental datasets have implicated increasing angular branch points in generating differential WSS patterns, across the vessel branching points. However, a majority of these reports fail at elaborating on the ramifications of differential WSS patterns caused by incremental angles of bifurcation, on endothelial dysfunction.

EC is the inner lining of the blood vessels that are more susceptible to changes in blood flow hemodynamics and WSS[11,12]. Indeed, EC dysfunction due to lower WSS at vessel branching points is considered one of the primary causes of the early atherosclerotic switch of the vascular bed[13]. EC dysfunction at blood vessel branching points is manifested by dramatic changes in the expression level of certain proteins responsible for the regulation of endothelial inflammation. For instance, EC at the vasculature branching point exhibits a lower expression level of endothelial pro-survival genes such as eNOS, KLF2 and KLF4 while displaying a greater expression of inflammatory adhesion molecules, including ICAM1, VCAM1 and P-selectin[14]. Such abrupt changes in endothelial gene/protein expression predispose these cells to allow attachment and activation of inflammatory cells, including monocytes and leucocytes, supporting atherosclerosis development[15,16]. Because endothelial dysfunction is directly correlated with the extent of lower WSS pattern, bifurcation angles of blood vessels greatly influence the inflammatory state of EC, including altering the physiological functioning of the vascular bed. Thus, the extent of endothelial inflammation is considered a hallmark to quantify the hemodynamic changes, including oscillatory/turbulent flow and its associated lower WSS effect on the vascular wall. Through the present study, we evaluate the influence of the bifurcation angle of a branched human coronary artery in contributing toward hemodynamic and associated WSS changes, and further correlated such biomechanical alteration to that of endothelial inflammation, a primary cause of atherosclerosis onset and progression.

## Results and discussion

Blood vessel branch points have been well understood in generating pulsatile or oscillatory D-Flow patterns coupled with reduced WSS levels. Contemporary findings have identified outer walls of bifurcations as the regions experiencing D-Flow and have marked them as "athero-prone"[8]. In addition, evidence hints towards a likely involvement of vessel bifurcation in vascular afflictions[10]. Indeed, a patient-specific study to evaluate the diagnostic performance of left coronary bifurcation angles and plaque characteristics for the prediction of coronary stenosis showed that a wider bifurcation angle of the coronary artery was associated with significant left stenosis, greater plaque burden, and non-calcified plaques[17]. We, therefore, set out to determine the differential effects of the varying degrees of bifurcation angles on fluid flow, WSS pattern, and endothelial upkeep. In so doing, we numerically modelled the branched human right-coronary artery (Supplemental Fig. 1a, b) with bifurcation angles of 30°, 40°, 50°, 60°, 70° and 80°. CFD analysis using a steady-state flow, revealed a unique pattern of WSS at the bifurcations: an observable decrease in the WSS levels (D-Flow) with an increase in the angle of bifurcation (Fig. 1a–g). While there was no significant change in the D-Flow WSS level for a 30° bifurcation (Fig. 1a, g), the 40° branch point displayed a modest reduction (Fig. 1b, g). We noted a further decrease in the 50° D-Flow WSS level (Fig. 1c, g), with the 60° bifurcation showing the least WSS levels (Fig. 1d, g). Interestingly, no further reduction in the WSS was reported in the case of 70 and 80° bifurcations with levels at par with 50 and/or 60° (Fig. 1e–g). Moreover, we observed that the area of low WSS in the bifurcation region increases with an increase in the branching point angle. However, in vivo blood circulation is indeed pulsatile as opposed to steady-state. Thus, we wondered if a pulsatile flow would lead to similar D-Flow WSS levels, with varying angles of bifurcation. The time average WSS (TAWSS) of the pulsatile simulation studies, although not as detrimental as steady-state, followed a similar trend (Fig. 1g–m). We reported vessels with 50°, 60° and 80° bifurcations caused a significant reduction in TAWSS, with the 60° branching exhibiting the most robust effect (Fig. 1g–m). Furthermore, to simulate a highly realistic scenario, we implemented fluid-structure interaction (FSI) model for CFD analysis using pulsatile flow. Through such analysis, we reported that pulsatile-FSI simulations displayed lower D-Flow shear levels (0.2–0.8 dyn/cm$^2$) as compared to pulsatile flow with rigid boundary conditions (Fig. 1g). Also, as anticipated, pulsatile-FSI simulations reduced D-Flow WSS levels with increasing angular branching points.

Over the years, researchers from across the world have emphasized the requirement of an optimal WSS for vascular homeostasis. Disturbances in the steady laminar flow caused at the vessel branch points often manifest endothelial inflammatory, senescent, and apoptotic phenotypes. Lowered WSS and oscillatory shear stress were both identified to be crucial in plaque formation, including the buildup of larger lesions with a vulnerable plaque phenotype, whereas vortices with oscillatory shear stress induce stable lesions[18]. The functional regulation of the endothelium by local hemodynamic WSS was found to be an ideal indicator of the focal propensity of atherosclerosis in the setting of systemic factors and may help guide future therapeutic strategies[19]. Precursory findings of our group illustrate the role of epigenetic mechanisms in regulating endothelial inflammation and apoptosis at bifurcations[20]. Therefore, we wondered whether a differential change in the WSS levels that corresponds to the increasing angle of vessel bifurcations would proffer a similar effect on the endothelial phenotype. To do so, we first 3D-printed the human right-coronary artery modelled with varying angles of bifurcation (30–80°) (Supplemental Fig. 1c). Using these channels, we then cast PDMS moulds for in vitro experimentations

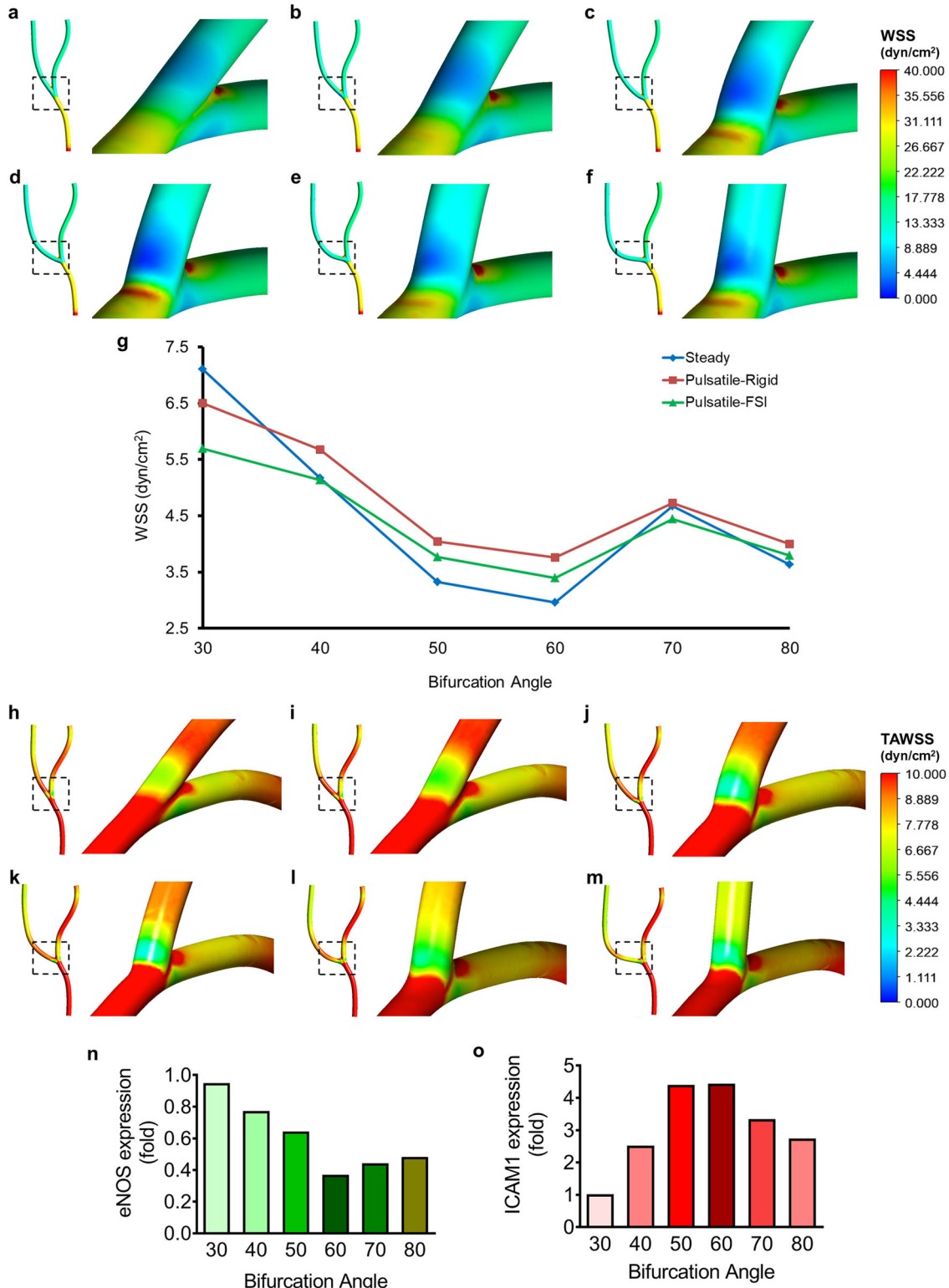

**Fig. 1 Increasing angular bifurcations of coronary artery creates regions with low WSS and alters biochemical signatures of EC experiencing D-Flow.**
**a**–**f** WSS contours of steady-state flow simulations in the human coronary artery with bifurcation angles of 30° (**a**), 40° (**b**), 50° (**c**), 60° (**d**), 70° (**e**) and 80° (**f**). **g** WSS or TAWSS levels under steady-state, pulsatile-rigid and pulsatile-FSI conditions in the D-Flow regions (outer walls of vessel branching point) 30, 40, 50, 60 and 80° bifurcations. **h**–**m** TAWSS contours of pulsatile-rigid studies in the bifurcation region of branched coronary artery with angles 30° (**h**), 40° (**i**), 50° (**j**), 60° (**k**), 70° (**l**) and 80° (**m**). **n**, **o** Alterations in the protein level expressions eNOS (**n**) and ICAM1 (**o**), with varying angles of vessel bifurcations, in EA.hy926 exposed to 4 h flow in the 30, 40, 50, 60, 70 and 80° microchannels.

(Supplemental Fig. 1c). Moreover, a sandwich of such vessel harbouring mould and gelatin-coated glass slide was used to grow and expose EC to fluid shear stress (Supplemental Fig. 1d, e), followed by immunofluorescence-based assessment of eNOS and ICAM1 expression levels. Through such analysis, we obtained a unique trend in the protein level expressions of eNOS and ICAM1, which corresponded with the differential reduction in WSS across varying degrees of bifurcations. eNOS being a marker of a healthy endothelium, showed a gradual reduction at vessel branch points, especially in the variable arm, with very little to no change at 30°, maximum decrease at 60° followed by an equivalent reduction in 70 and 80° in comparison to their respective laminar flow (S-Flow) exposed areas (Fig. 1n). Whereas in case of ICAM1 (a marker of endothelial inflammation), we reported a substantial increase in its protein level expression starting from 30°, with the highest expressions at 50 and 60° and a marginally lesser expression at 70 and 80° bifurcations (Fig. 1o). To elaborate, no significant changes in the eNOS (Supplemental Fig. 2a) and ICAM1 (Supplemental Fig. 2b) expressions were observed in case of a 30° bifurcation. As for the 40° bifurcation, 22% reductions in the eNOS (Supplemental Fig. 2c) while a 60% increase in the ICAM1 (Supplemental Fig. 2d) expressions were reported. We noted a 35% lower and a 77% increase in eNOS and ICAM1 levels (Supplemental Fig. 2e, f), respectively, for the 50° bifurcation. Surprisingly, in the 60° mould, while we evaluated a massive 63% drop in eNOS at the branch point (Supplemental Fig. 2g), only a 77.5% increment in the ICAM1 level was evident (Supplemental Fig. 2h). Staying true to the aforementioned trend, eNOS expression levels plummeted by just 56% and 51% in 70 and 80° angled bifurcations (Supplemental Fig. 2i, k), respectively. In parallel, the ICAM1 expressions in the EC at 70 and 80° bifurcations surged by 70% and 63% (Supplemental Fig. 2j, l), respectively.

Blood as a whole is a complex suspension with corroborated non-Newtonian rheological characteristics[21]. Nonetheless, the concurrent finding suggests that the overall velocity and wall pressure distributions remain similar in the case of both non-Newtonian as well as Newtonian fluid models[22]. We, therefore, set out to determine the velocity dispensation across the bifurcations with varying angles. In so doing, we generated velocity vectors of the fluid flow through the modelled coronary artery with branch point and encountered a remarkable distribution pattern. However, firstly, as described earlier, the bifurcated coronary artery was modelled so that the right arm/outlet remains at a fixed angle with the inlet stem/arm, while the left arm/outlet is at variable angles with the former (Supplemental Fig. 1c). As for the distinct pattern, the velocity in the left/variable arm gradually decreased with a consequently equal rise in the right/fixed arm, in parallel with increasing bifurcation angle from 30° to 80° (Fig. 2a–g)—in case of steady-state flow. Moreover, an increase in the bifurcation angles can be characterized by the boundary layer separation, diminishing fluid velocities and elevated levels of turbulence (crisscross/rebound vectors) at the outer walls (Fig. 2a–f). Simulations with pulsatile flow showed similar velocity dispensation across the two outlets along with enhanced eddy formations and recirculations near the outer walls —with increasing bifurcation angles (Fig. 2h–m). Although the size of the vortices increases from 30 to 80°, 60° branching point exhibits the lowest TAWSS because of the larger wake zone (Supplemental Fig. 3a–f). For 70 and 80° bifurcation angles, the prominent recirculations reduce the wake zone, thereby increasing the TAWSS levels. In addition, it is well understood that the direction of the shear stress vector is determined by the direction of the velocity vector[23]. Therefore, we closely monitored the WSS levels obtained through CFD analysis. Interestingly, we observed differences between WSS/TAWSS levels of the two outer walls of

bifurcation, viz., right/fixed arm bifurcation (D-Fixed), and left/variable arm bifurcation (D-Variable) in steady-state as well as pulsatile fluid flow. Such difference in the case of the 30° bifurcation was the least gradually increasing to reach a maximum at 60° with differences in 70 and 80° bifurcations not far behind (Fig. 2n). Furthermore, we also reported a similar trend in the WSS/TAWSS levels of the two outlet arms (post bifurcation) with increasing angles of bifurcation (Fig. 1a–f, h–m). Surprisingly, an increase in the WSS just before the branching point was reported with increasing angular bifurcations for both steady-state and pulsatile flow. Such phenomena are the result of truncating fluid volumes flowing through the variable arm/outlet with increasing bifurcation angles. Additionally, these varying fluid volumes flowing across the two outlets lead to the fluid spending more time in the D-Flow regions prior to entering the fixed arm/outlet resulting in a higher oscillatory shear index (OSI) (Supplemental Fig. 3g–l) and relative residence time (RRT) (Supplemental Fig. 3m–r). The recirculations at the vessel bifurcation cause a directional change in the WSS, consequently increasing the OSI value. Apart from the ranching point, regions of the artery with curvature are also marked by higher OSI and increased RRT. Thus, the probability of plaque formation at D-Variable is higher than that at D-Fixed.

With these novel insights learned from the numerical simulation, we wondered whether such a differential WSS pattern over a range of bifurcation angles realistically occurs in the biological system and, if so, what would be its implications. To cater to this thought, we specifically chose three different angles, 30° (no significant changes in WSS and velocity distribution at bifurcation), 60° (maximum reduction in WSS with a notably different velocity distribution at bifurcation) and 80° (sizable reduction in WSS with a highly disturbed velocity dispensation at the bifurcation); and began by observing the fluid flow pattern within the respective microchannels. To do so, we circulated magnetic beads suspended in phosphate-buffered saline through the microchannels and recorded how the beads behaved at the vessel bifurcation. We observed no flow pattern changes in the case of the 30° branching point, while a significant slow-down of the magnetic beads was reported in both 60 and 80° microchannels, especially in D-variable regions (Supplemental Videos 1–7). We next performed in vitro area-specific assessment of the endothelial phenotype. In doing so, we cultured EC in the respective moulds exposed them to 4 h of fluid flow and carried out the immunofluorescence-based analysis of eNOS and ICAM1 protein expression levels. We started by identifying and defining different zones of the modelled coronary artery as, inlet arm: S-Flow, right/fixed bifurcation: D-Fixed, left/variable bifurcation: D-Variable, right/fixed outlet arm: S-Fixed, and left/variable outlet arm: S-Variable (Fig. 2o). For the 30° channel, we reported no significant changes in the eNOS (Fig. 3a and Supplemental Fig. 4a) expression levels across various locations. In contrast, we observed a significant reduction in the eNOS level at the 60° bifurcation point with no difference in the eNOS levels was reported either between D-Fixed and D-Variable or between S-Fixed and S-Variable regions (Fig. 3b and Supplemental Fig. 4a). In case of the 80° channel, significant reduction in eNOS expression levels were evident including a pronounced difference in eNOS level was recorded among D-Fixed and D-Variable, and S-Fixed and S-Variable regions (Fig. 3c and Supplemental Fig. 4a). To better visualize such area-specific difference in the expression levels of eNOS across the microchannel, we performed tile scan imaging of EC stained for eNOS exposed to fluid flow in microchannels containing 30°, 60° and 80° bifurcation angels and observed the previously elaborated differences in eNOS level across the microchannels as reported with higher magnification images (Fig. 3d). Further spatial analysis of ICAM1 in 30°, 60°

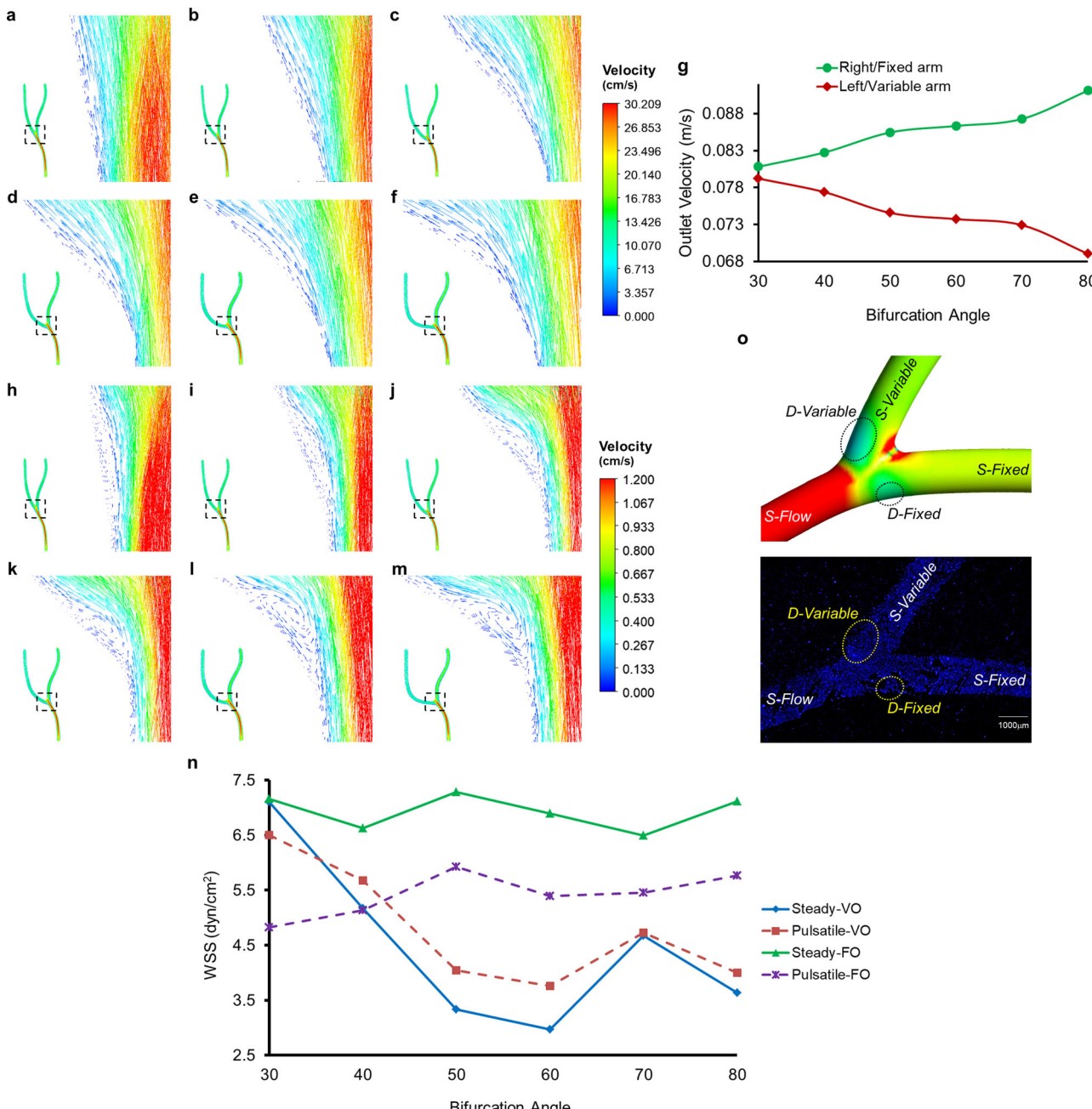

**Fig. 2 Angular differences at the coronary artery branching points cause turbulent flow patterns, perturb velocity dispensation thereby forming WSS gradient. a–f** Velocity vectors on steady-state simulations of 30° (**a**), 40° (**b**), 50° (**c**), 60° (**d**), 70° (**e**) and 80° (**f**) bifurcation angles. **g** Outlet fluid flow velocities (m/s) obtained from right/fixed arm and left/variable arm of human coronary artery with 30, 40, 50, 60, 70 and 80° branching points. **h–m** Velocity vectors of pulsatile-rigid simulations in coronary artery with bifurcation angle 30° (**h**), 40° (**i**), 50° (**j**), 60° (**k**), 70° (**l**), and 80° (**m**) at t = 0.415 s. **n** WSS or TAWSS in the D-Flow regions variable and fixed for steady-state and pulsatile-rigid simulations. **o** Based on the simulation studies performed on a bifurcated human right-coronary artery, various zone were identified and defined across the vessel. Inlet arm: steady laminar flow (S-Flow), right/fixed outer wall (bifurcation): disturbed flow in fixed arm (D-Fixed), left/variable outer wall (bifurcation): disturbed flow in variable arm (D-Variable), steady laminar flow outlet arms: right/fixed arm (S-Fixed) and left/variable arm (S-Variable). Similar regions were identified and used for in vitro microscopic analysis EC exposed to shear stress in the microchannels.

and 80° microchannels, we were unable to detect any immunostaining of ICAM1 with in the 30° channel (Fig. 4a and Supplemental Fig. 4b) while microchannel with 60° bifurcation angel exhibited robust increase in the ICAM1 level in D-flow regions while no difference in the levels of ICAM1 was detected either between D-Fixed and D-Variable or between S-Fixed and S-Variable regions (Fig. 4b and Supplemental Fig. 4b). In case of the 80° channel, significant gain in ICAM1 expression levels were

evident at branching points with pronounced differences in ICAM1 expressions were observed among D-Fixed and D-Variable, and S-Fixed and S-Variable regions (Fig. 4c and Supplemental Fig. 4b). Similar to eNOS staining, we performed tile scan imaging of ICAM1 stained EC which were exposed to shear stress in 30°, 60° and 80° microchannels and recorded similar differences as reported with higher magnification images (Fig. 4d). To further corroborate the pro-atherogenic EC phenotype we

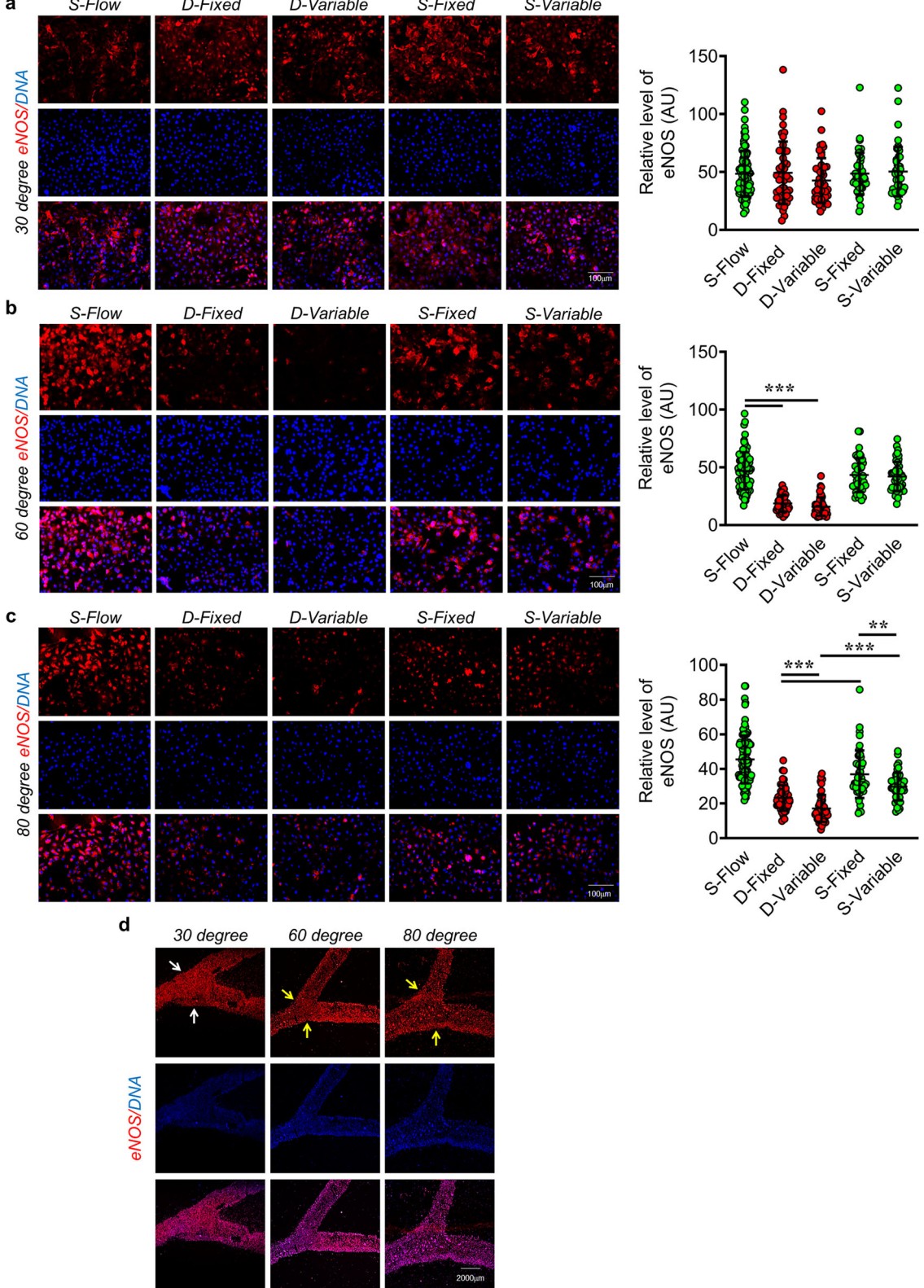

**Fig. 3 Exposure of EC to D-Flow in microchannels with varying angles of bifurcation leads to reduced eNOS expressions. a–c** Immunofluorescence staining of EA.hy926 for eNOS ($n = 3$) after 4 h fluid flow exposure, using human coronary microchannels with 30° (**a**), 60° (**b**) and 80° (**c**) branching points. DAPI staining is shown in blue. eNOS fluorescence signal in individual EA.hy926 cells (dots) from three individual experiments. Fluorescence intensity AU values per individual cells are indicated together with the mean. Total number of cells, $n \geq 45$. Values represent the mean ± SD. *$p < 0.05$, **$p < 0.01$ and ***$p < 0.001$, by one-way ANOVA. Magnification: ×20, Scale: 100 μm. **d** Tile scan images microchannels harbouring EC stained for eNOS. Magnification: ×10, Scale: 2000 μm.

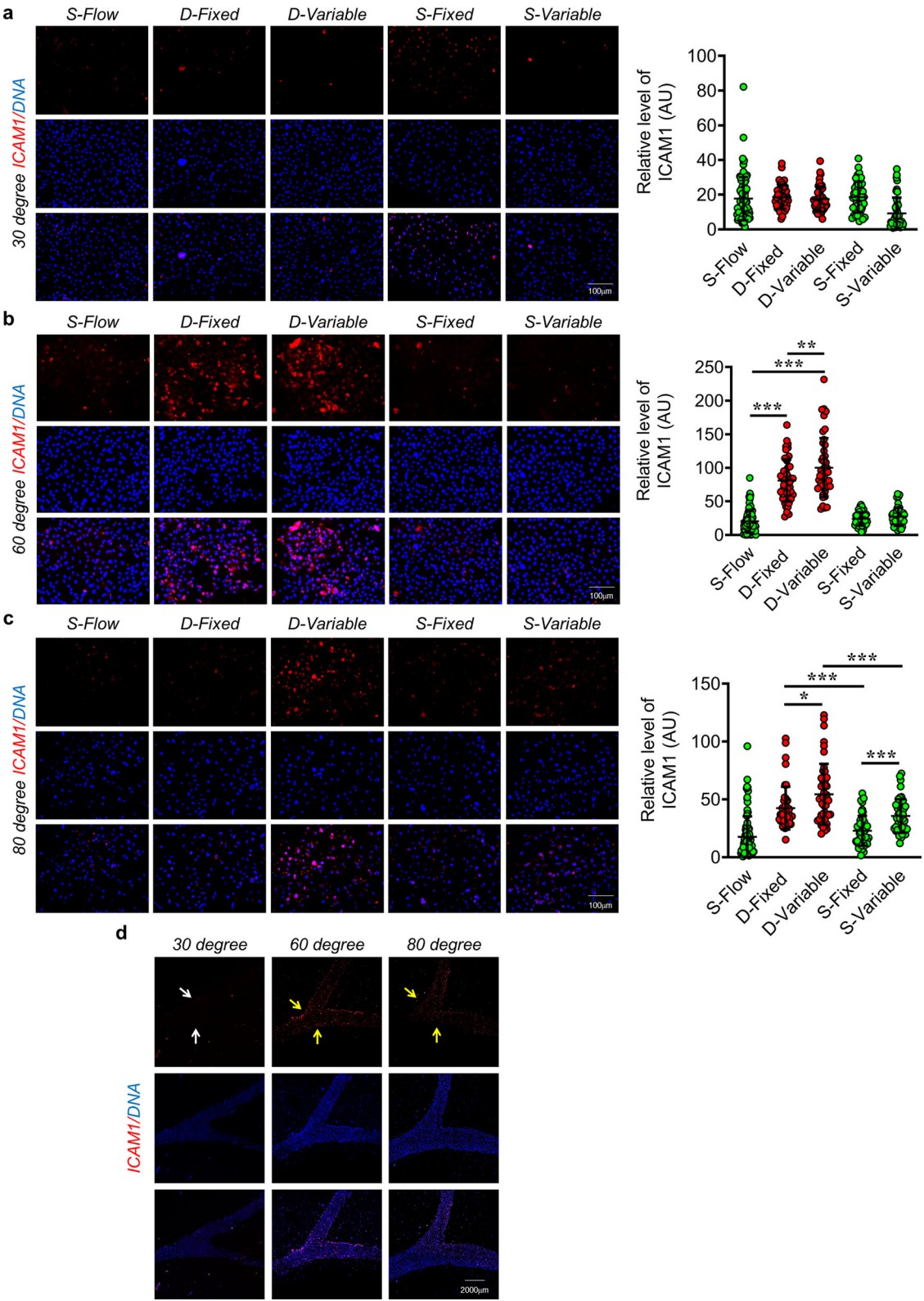

**Fig. 4 D-Flow exposure causes elevated expression of pro-inflammatory ICAM1 in EC. a–c** Immunofluorescence staining of EA.hy926 for ICAM1 ($n = 3$) after 4 h fluid flow exposure, using human coronary microchannels with 30° (**a**), 60° (**b**) and 80° (**c**) branching points. DAPI staining is shown in blue. ICAM1 fluorescence signal in individual EA.hy926 cells (dots) from three individual experiments. Fluorescence intensity AU values per individual cells are indicated together with the mean. Total number of cells, $n \geq 45$. Values represent the mean ± SD. *$p < 0.05$, **$p < 0.01$ and ***$p < 0.001$, by one-way ANOVA. Magnification: ×20, Scale: 100 μm. **d** Tile scan images microchannels harbouring EC stained for ICAM1. Magnification: ×10, Scale: 2000 μm.

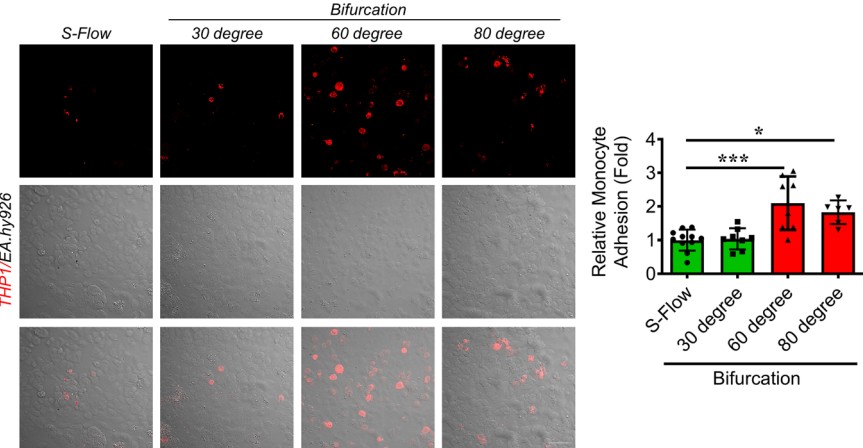

**Fig. 5 Varying angles of vessel bifurcation induce pro-atherogenic-endothelial dysfunction.** Monocyte adhesion assay was performed to assess the impact of increasing angular bifurcations on EC function. Pre-stained THP-1 (human monocytes) were incubated with flow exposed (4 h) EA.hy926 for 30 mins. Adherent THP-1 (red dots) were counted in S-Flow and D-Flow regions of 30°, 60° and 80° microchannel.

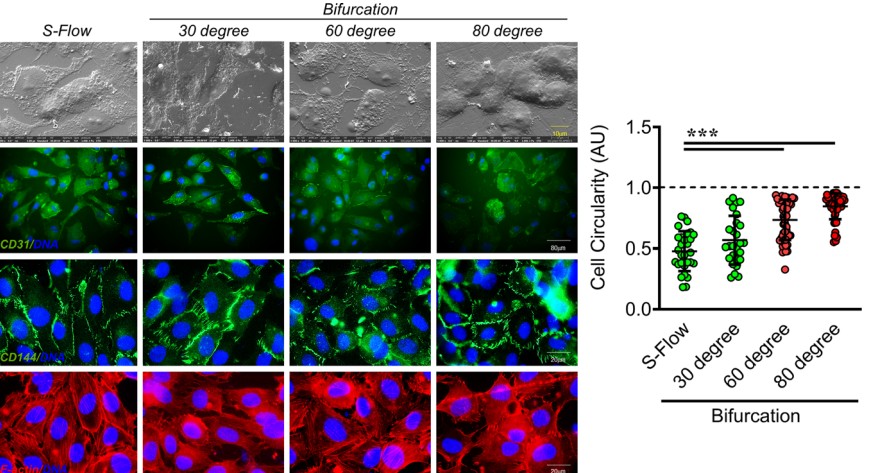

**Fig. 6 Increasing bifurcation angles of coronary artery incrementally impacts the EC morphology.** FE-SEM (top) based structural and surface level analysis of EC exposed to D-Flow using 30°, 60° and 80° microchannels. Magnification: ×5000, Scale: 10 μm. Flow exposed (4 h; 30°, 60° and 80° microchannels) EA.hy926 cells stained for cell-cell contact markers CD31 (second) and CD144 (penultimate). DAPI staining is shown in blue. Circularity of such cells was assessed using ImageJ. Cell circularity AU values per individual cells are indicated together with mean. Total number of cells, $n \geq 30$. Values represent the mean ± SD. *$p < 0.05$, **$p < 0.01$ and ***$p < 0.001$, by one-way ANOVA. Magnification: ×50/×63, Scale: 80 μm/20 μm. Cell cytoskeletal F-actin (bottom) staining for EA.hy926 exposed to D-Flow (D-variable or D-fixed regions) in 30°, 60° and 80° microchannels, using phalloidin. DAPI staining is shown in blue. Magnification: ×63, Scale: 20 μm.

performed monocyte adhesion assay. Monocyte adhesion to the endothelium during inflammation and its association with atherosclerotic lesions has long been recognized[24,25]. In comparison to S-flow, we reported a 2-fold and a 1.6-fold increase in the monocyte adhesion at the 60° and 80° vessel branching points, respectively, while no significant changes were observed in the case of the 30° bifurcations (Fig. 5).

WSS is a well-established determinant of EC function, gene expression, and its structure[26,27]. Having observed a unique pattern in the area-specific eNOS and ICAM1 expressions, and differential monocyte adhesion potential (results of differential WSS) with increasing bifurcation angles, we were curious in finding if such a differential WSS pattern had a similar effect on EC morphology. To address this, we performed Field Emission-Scanning Electron Microscopy (FE-SEM), immunofluorescence-based assay for cell circularity, and F-actin staining. Through FE-SEM imaging, we reported long and inflated cells in S-Flow and 30° bifurcation, flat and round morphology at the 60° bifurcation,

and inflated but circular cellular structure in the 80° bifurcation region (Fig. 6 top panel, Supplemental Fig. 5a). Similarly, immunofluorescence staining for CD31 and CD144 (Fig. 6), and F-actin staining with phalloidin revealed elongated cells in S-Flow and 30° bifurcation, a mix of long and round cells in 60° and majorly circular cells in 80°(for bifurcation either D-variable or D-fixed regions are considered, Fig. 6 and Supplemental Fig. 5b). Previous work by Liu et al. studied the influence of different coronary bifurcation angles on the blood flow field and related hemodynamic parameters through in silico approach. The authors also further predicted whether such hemodynamic alterations due to changes in bifurcation angle also allow understanding of the susceptibility of such blood vessels to develop atherosclerotic plaques. A complex CFD analysis using computer-modelled blood vessels with different bifurcation angles revealed that a wider bifurcation angle can cause a wider low-wall shear stress area, leading to atherosclerosis. In contrast, a decreased angle between the branched blood vessels prevents

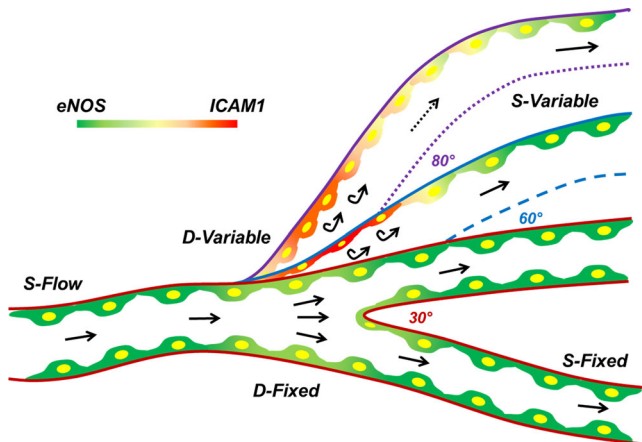

**Fig. 7 Differential angular branching points of a human coronary artery cause varying degrees of fluid flow turbulence consequently switching EC towards pro-atherogenic phenotypes.** Vessel branch points are known to affect the WSS levels, cause endothelial dysfunction and thus are the prime spots of atherosclerotic plaque formation. Increasing angles of the branching points differentially impact the balance of genes responsible for EC homeostasis. A bifurcation angle of 30° exhibits no significant alterations in the WSS levels, velocity distribution and protein level expressions of eNOS and ICAM1. On the contrary, vessel branching at an angle of 60° displayed the lowest WSS levels at the outer walls of the bifurcation. EC in the D-Flow regions of such a vessel were found to have a round and flat morphology with high levels of ICAM1 and a concurrent reduction in eNOS expression, in comparison to S-Flow. The larger angular disturbance caused by the 80° branching, affected WSS to similar levels; however, a huge alteration was observed in the velocity dispensation across the bifurcation. These differential velocity gradients are responsible for generating special zones with varying WSS levels and a corresponding change in eNOS and ICAM1 expression.

atherosclerosis[28]. Furthermore, a clinical study involving patients with coronary artery diseases, also indicated that coronary artery having a bifurcation angle of 80.5° and above are 84.1% sensitive and 81.3% specific in developing an atherosclerotic lesion localized in 5 mm length from the point of bifurcation site. The author concluded that as the bifurcation angle increases, atherosclerotic lesions tend to approach the bifurcation site and suggested that lesions with increased angulation may need extra care as they are more likely to present with further complications[7]. This is in parallel to the observation reported in our study describing bifurcation angles of 60° and above are more vulnerable to generate complex hemodynamics changes that will support more robust induction of endothelial dysfunction and inflammation, which is one of the hallmarks associated with atherosclerosis onset and progression.

In summary, through in silico and in vitro approaches, we reported that blood vessels with varying angles of bifurcation exhibit a differential effect on the WSS levels and the resulting endothelial phenotype. A bifurcation angle of not more than 40° manifests minimal to no consequences on the vasculature. Furthermore, branch points with angles between 50° and 60° have detrimental effects on the endothelial lining. Vessel bifurcations with angles in excess of 70° display a moderate to appalling effect on the EC (Fig. 7). Although the current study reported several new findings and introduced 3D-printed human coronary artery model system for cellular response studies, we are still far from simulating the real blood flow conditions and its effect on cellular response in vivo. For instance, blood is a more complex media containing different cells, including cells of the immune origin

**Table 1 Mesh properties: element count and maximum skewness for each geometry.**

| Angle of geometry (°) | Solid domain | | Fluid domain | |
|---|---|---|---|---|
| | Elements | Maximum skewness | Elements | Maximum skewness |
| 30 | 39,996 | 0.89763 | 472,644 | 0.88350 |
| 40 | 41,277 | 0.73581 | 510,856 | 0.83234 |
| 50 | 42,277 | 0.77345 | 494,903 | 0.77815 |
| 60 | 42,836 | 0.59102 | 489,119 | 0.75532 |
| 70 | 42,703 | 0.61489 | 487,903 | 0.75220 |
| 80 | 43,945 | 0.60617 | 506,420 | 0.74995 |

which roles over the EC layer during blood flow. Such phenomenon in in vivo settings likely governs the response of endothelial monolayer to fluid flow, which is indeed not addressed through the current model. In addition, due to limitations in the capability of our peristaltic pump, we are unable to generate in vivo a pulsatile flow, which could have a more robust effect on endothelial response to differential flow due to angular differences in bifurcation. Nonetheless, to the best of our knowledge, to date, no study amalgamated these two analyses and observations together to achieve the goal of the present study; the correlation of CFD-based analysis with the biological response at the cellular level through cell culture-based experimentation. In an array of multiple pre-existing studies that have focused on either the computational or the biological aspects of pro-atherogenic-endothelial dysfunction, however, the current study presents an intriguing amalgamation of the two. Herein, we put forth a detailed account of CFD simulations, including steady-state, pulsatile, and pulsatile-FSI flow studies and endorsed these in silico findings with in vitro functional, biochemical, and morphometric evaluations. Outcomes of the study can assist clinicians in atherosclerosis risk-assessment prediction from 3D-angiograms/CT scans, and may also prove valuable to surgeons in performing a coronary bypass.

## Methods

**Modelling an exemplar human coronary artery.** CT scan images of the right human coronary artery with a bifurcation were obtained from the SimVascular web portal (https://simvascular.github.io/). A three-dimensional (3D) artery with a total length of approximately 6 cm, a diameter of 2 mm and a distance of 2 cm between two arms was modelled using Fusion 360 (AutoDesk®) (Supplemental Fig. 1a). Furthermore, beginning with 30° to a maximum of 80° the angle between the branching/outlet arms was varied in increments of 10°. Herein, to make a fair assessment, the angle between the inlet arm and one of the outlet arms was fixed while changing the angle of bifurcation[29]. The spline tool helped obtain the curvature of the arteries. In addition, to overcome the edge effect, a fillet was employed for smoothening out the edges at the bifurcation. Standard Tessellation Language (STL) files were created using free computer-aided design (CAD) software and imported to ANSYS Workbench for mesh generation. A tetrahedron meshing for the fluid domain (0.25 mm element size) and quadrilateral meshing for the solid domain (0.15 mm element size) was applied along with the patch conforming algorithm. At the bifurcation, meshing was refined using the sphere of influence with an element of 0.15 mm and 0.1 mm for fluid and solid domains with 11.5 mm radius, respectively (Table 1). For better accuracy at the solid-fluid interface, prismatic layers were generated with the post-inflation algorithm and smooth transition with a ratio of 0.272, 6 layers and 1.2 as growth rates.

**Computational fluid dynamics (CFD) simulation**

*Blood viscosity model.* We considered the blood as a non-Newtonian fluid because of plasma, red blood cells, white blood cells, and other particles present in it. This non-Newtonian shear-thinning nature of the blood is important in the bifurcation region and stenosis studies. Several viscosity models such as Carreau, Carreau-Yasuda, Power-law, Casson, modified-Casson, and Walburn-Schneck models are available. However, the modified-Casson model is reported in the literature to provide accurate information by yield stress and the shear-thinning nature of the blood in the vessel with a smaller diameter[30,31]. The

modified-Casson model is represented by[32],

$$\mu = \left( \sqrt{\mu_c} + \frac{\sqrt{\tau_0}}{\sqrt{\lambda} + \sqrt{\psi}} \right)^2 \tag{1}$$

where $\mu_c$ ($= 4 \times 10^{-3}$ Pa.s) is Casson viscosity, $\tau_0$ ($=0.021$ Pa) is yield stress, $\psi$ is the rate of deformation of fluid, and $\lambda$ ($=11.5\,s^{-1}$) is a constant introduced to have viscosity even when the rate of deformation of fluid is zero. The density of blood was considered to be 1060 kg/m$^3$.

*Flow model.* Blood flow in the arteries was modelled by solving the continuity equation and Navier-Stokes equation, which gives information about the conservation of momentum[33]. These equations were numerically solved simultaneously in the academic version of ANSYS Fluent software, version 17.2.

The mass and momentum continuity equations are expressed below,

$$\frac{\partial \rho_f \vec{v}}{\partial t} + \nabla \cdot \rho_f \vec{v} = 0 \tag{2}$$

$$\frac{\partial \rho_f \vec{v}}{\partial t} + \nabla \cdot (\rho_f \vec{v}\vec{v}) = -\nabla p + \nabla \cdot \bar{\bar{\tau}} \tag{3}$$

where $\vec{v}$ is a flow-velocity vector; $\rho_f$ is the density of blood; $p$ is the pressure, and $\bar{\bar{\tau}}$ is viscous stress tensor expressed as $\bar{\bar{\tau}} = \mu[\nabla\vec{v} + (\nabla\vec{v})^T]$, ignoring the volume dilation. Here $\mu$ is the viscosity of the blood represented by the modified-Casson model. The first term represents the time-dependent flow that can be neglected for steady-state simulations.

We used a laminar flow model as a base model to perform the steady-state simulations. However, we observed backflow and eddy formation in the bifurcation region. Therefore, we used the SST $k$-$\omega$ model for pulsatile studies[34], which provides better accuracy when the fluid flow is in the low-Re turbulence and the viscous-sublayer region.

*Shear stress transport* k-$\omega$ *model.* Shear Stress Transport (SST) $k$-$\omega$ model combines the Wilcox $k$-$\omega$ model and $k$-$\varepsilon$ model to predict outcomes accurately in the area near the wall and free stream. The $\omega$ equation in the SST model includes a damped cross-diffusion derivative element. The $k$ equation is modified to account for the transport of the turbulent shear stress. These modifications in $k$ and $\omega$ equations provide relevant results in both near-wall and far-field zones. The equations[35] are expressed as follows.

$$\frac{\partial}{\partial t}(\rho_f k) + \frac{\partial}{\partial x_i}(\rho_f k u_i) = \frac{\partial}{\partial x_j}\left(\Gamma_k \frac{\partial k}{\partial x_j}\right) + \widetilde{G}_k - Y_k + S_k \tag{4}$$

$$\frac{\partial}{\partial t}(\rho_f \omega) + \frac{\partial}{\partial x_i}(\rho_f \omega u_i) = \frac{\partial}{\partial x_j}\left(\Gamma_\omega \frac{\partial \omega}{\partial x_j}\right) + G_\omega - Y_\omega + D_w + S_\omega \tag{5}$$

where $\rho_f$ represents fluid density, $k$ represents the turbulent kinetic energy, $\omega$ represents the specific dissipation rate, $\widetilde{G}_k$ represents the generation of turbulent kinetic energy due to mean velocity gradients, $G_\omega$ represents the generation of specific dissipation rate, $\Gamma_k$ and $\Gamma_\omega$ represent the effective diffusivity of $k$ and $\omega$ respectively, $Y_k$ and $Y_\omega$ represents the dissipation of $k$ and $\omega$ due to turbulence, $D_w$ represents the cross-diffusion term, and $S_k$ and $S_\omega$ represent the user-defined source terms.

*Fluid-structure interaction (FSI) model.* As the arterial wall is elastic, the forces acting on the arterial wall due to fluid flow distorts the arterial geometry, and the distortion has a significant impact on the fluid flow and the WSS acting on the wall. This phenomenon is described by the FSI model.

*Solid model.* The equation that governs the elastic nature of the wall is developed from Newton's second law of motion and is mathematically defined by[30,36],

$$\rho_s \frac{\partial^2 \varepsilon}{\partial t^2} = \nabla \sigma_s + \rho_s F \tag{6}$$

where $\rho_s$ is the density of the artery wall having a value of 1120 kg/m$^3$. $\varepsilon$ and $\sigma_s$ correspond to the displacement and stress component of the arterial wall, respectively. $F$ is the body forces acting on the arterial wall. The value of $\sigma_s$ is found from the material properties of the arterial wall[37] the arterial wall is considered incompressible and hyper-elastic. The hyper-elastic nature of the wall is described by the 9-parameters Mooney–Rivlin model[30] and expressed in terms of strain-energy density function ($W$), which measures the energy accumulated in the process of the distortion, is given by Eq. (7).

$$\begin{aligned} W = {} & c_{10}(\bar{I}_1 - 3) + c_{01}(\bar{I}_2 - 3) + c_{20}(\bar{I}_1 - 3)^2 + c_{11}(\bar{I}_1 - 3)(\bar{I}_2 - 3) \\ & + c_{02}(\bar{I}_2 - 3)^2 + c_{30}(\bar{I}_1 - 3)^3 + c_{21}(\bar{I}_1 - 3)^2(\bar{I}_2 - 3) \\ & + c_{12}(\bar{I}_1 - 3)(\bar{I}_2 - 3)^2 + c_{03}(\bar{I}_2 - 3)^3 + \frac{1}{D_1}(J - 1)^2 \end{aligned} \tag{7}$$

where $C_1 = 0.070$, $C_3 = 3.2$, $C_7 = 0.07160$ and $C_i = 0.0 (i = 2, 4, 5, 6, 8 \text{ and } 9)$ MPa are called as Mooney–Rivlin constants and $d$ ($= 10^{-5} Pa^{-1}$) is the material incompressibility factor, $J$ is the ratio of deformed elastic volume to inconvertible volume and $\bar{I}_1$, $\bar{I}_2$ and $\bar{I}_3$ are first, second, and third deviatoric strain invariants, respectively. The values of the Mooney–Rivlin parameters are obtained from the experiments made on healthy human coronary arteries[38].

*Boundary conditions.* Simulation studies were carried out with steady and pulsatile boundary conditions compared to experimental studies.

For steady simulations, the blood was pumped at a constant inlet velocity of 0.16 m/s, and the outlets were left to atmospheric pressure. As the blood flow in the human body is pulsatile, a pulsatile velocity profile was considered in both the inlet (Eq. 8) and the outlet (Eq. 10) conditions[39]. A no-slip condition was applied to the flow near the wall, i.e., zero velocity at the fluid-solid interface.

The velocity profile is given by,

$$U(t) = Q(t)/A \tag{8}$$

where $U(t)$ is the pulsatile velocity profile, $Q(t)$ is the pulsatile flow rate, and $A$ is the cross-sectional area of the inlet. The flow rate and pressure equations were represented using the Fourier series as follows,

$$Q(t) = \bar{Q} + \sum_{n=1}^{4} \alpha_n^Q \cos(n\omega t) + \beta_n^Q \sin(n\omega t) \tag{9}$$

$$p(t) = \bar{p} + \sum_{n=1}^{4} \alpha_n^p \cos(n\omega t) + \beta_n^p \sin(n\omega t) \tag{10}$$

where $\bar{Q}$ is the mean volumetric flow rate, $\omega = \frac{2\pi}{T}$ is the angular frequency with $T = 0.8s$ and $\bar{p}$ is the mean pressure. Table 2 presents the values of the parameters used in the velocity and pressure waveform equations.

*Solution method.* The pressure-based solver was considered in Fluent software as it was suitable for incompressible fluids flowing at low velocities. Pressure-Velocity coupling algorithm with the SIMPLE scheme as the solution method with a residual convergence limit of $10^{-6}$ for steady simulations and $10^{-4}$ for pulsatile conditions were used. The equations were discretized in second-order schemes to minimize errors and provide better accuracy. FSI equations were solved simultaneously through system coupling. Simulations were run at a time step of 0.005 sec for rigid and 0.016 sec for FSI studies, respectively.

*Mesh dependency.* As the results greatly depend on the quality and quantity of the mesh, the mesh dependency test was performed on the 60° bifurcation angle geometry by varying the element count with steady-state boundary conditions. The WSS in the bifurcation region of the variable outlet was monitored for mesh dependency (Table 3). The percentage difference was calculated with respect to the most refined mesh and found to be less than 1% for both meshing. So, the mesh with 0.5 million elements was considered to reduce the computational time.

*Hemodynamic descriptors.* Hemodynamic descriptors are well described in the literature, and they predict the impact of fluid flow in vascular diseases. WSS is the major parameter that helps determine the regions prone to atherosclerosis. As it is found that regions with low WSS are more prone to atherosclerosis, whereas region with higher values results in thrombogenesis.

While studying the pulsatile nature of the blood flow, Time Averaged Wall Shear Stress (TAWSS) gives the time-averaged value of WSS throughout the

## Table 2 Parameters of velocity and pressure waveform equations.

| Artery Vessel | $n$ | $\alpha_n^Q$ | $\beta_n^Q$ | $\alpha_n^p$ | $\beta_n^p$ |
|---|---|---|---|---|---|
| $\bar{Q} = 0.1589$ l/min | 1 | 0.1007 | 0.0764 | −3.3107 | −2.2932 |
| $\bar{p} = 84.9722$ mmHg | 2 | −0.0034 | −0.0092 | −9.8639 | 8.0487 |
| | 3 | 0.0294 | 0.0337 | 3.0278 | 3.8009 |
| | 4 | 0.0195 | −0.0129 | 2.2476 | −3.2564 |

## Table 3 Details of the mesh dependency study.

| | Mesh 1 | Mesh 2 | Mesh 3 |
|---|---|---|---|
| Number of elements | 505,692 | 621,236 | 700,451 |
| WSS | 0.99593 | 1.00595 | 1.00425 |
| Percentage difference | 0.8319% | 0.16913% | — |

cardiac cycle (Eq. 11). TAWSS value of less than 0.4 Pa means high susceptibility to plaque formation[40].

$$\text{TAWSS } (s) = \frac{1}{T} \int_0^T |WSS(s,t)| dt \qquad (11)$$

In pulsatile flows, the direction of the WSS shear acting on the wall varies in the cardiac cycle. The Oscillatory Shear Index (OSI) (Eq. 12), gives this variation. OSI is a dimensionless value having a value of 0 if the WSS is in the same direction (0°) of flow and 0.5 if the flow is in the opposite direction (180°). A higher value of OSI means the region is more prone to plaque formation[41].

$$\text{OSI}(s) = 0.5 \left[ 1 - \left( \frac{\left| \int_0^T WSS(s,t)\, dt \right|}{\int_0^T |WSS(s,t)|\, dt} \right) \right] \qquad (12)$$

Another important parameter is Relative Residence Time (RRT), which defines the time stayed by the particle near the wall. The higher the value of RRT refers to the higher directional change in fluid flow and lower TAWSS.

$$\text{RRT}(s) = \frac{1}{(1 - 2 \cdot OSI) \cdot TAWSS} \qquad (13)$$

Here, $t$ represents the instant time; s represents the location on the artery wall and $T$ is the total time of the cardiac cycle.

**Materializing the exemplar human coronary artery for in vitro differential flow simulation**. CAD files of the modelled human coronary arteries with bifurcation angles ranging from 30°, 40°, 50°, 60°, 70° and 80° were 3D-printed at the Cortex 3D Printing Facility, Birla Institute of Technology and Science Pilani, Pilani Campus, India. These 3D templates were used to cast channel-bearing poly-dimethylsiloxane (PDMS) moulds as described earlier[20] (Supplemental Fig. 1c). PDMS casts were sterilized with 70% ethanol followed by a 1 h exposure to UV light, prior to their use in vitro. A sandwich of such moulds and gelatin-coated glass slides was used to culture cells, thereby mimicking in vivo blood vessel-like differential flow conditions.

**Cell culture and in vitro shear stress exposure**. EA.hy926 cells were purchased from ATCC, Manassas, USA (#CRL-2922) and cultured using Dulbecco's modified Eagle medium (DMEM; Hi Media Laboratories, Mumbai, India) supplemented with 10% foetal bovine serum (FBS; Hi Media) and 1% penicillin–streptomycin (Sigma–Aldrich, Bangalore, India). Human umbilical vein endothelial cells (HUVEC) were procured from Hi Media (#CL002), and cultured using their Endothelial Cell Expansion Medium (#AL517) supplemented with 3% FBS and 1% penicillin–streptomycin. THP-1 human monocytes, a kind gift from Prof. Anil Jindal (Department of Pharmacy, BITS Pilani, Pilani Campus, India), were grown using the RPMI-1640 (#AL028A, Hi Media) supplemented with 10% FBS and 1% penicillin–streptomycin. All the cells were maintained in a humidified $CO_2$ incubator at 37 °C.

As for in vitro shear stress exposure, EC ($6 \times 10^5$) were seeded into the cast-slide sandwich. After a 4 h incubation (or until the cells settle and regain their morphology, whichever first), this sandwich was connected to the flow setup (Supplemental Fig. 1d). Regular growth medium was circulated using the MasterFlex® C/L® Analog Variable-Speed Pump Systems (Cole-Parmer, India) for a period of 4 h. Data from the fluid dynamics simulations helped define the regions experiencing steady laminar flow (S-Flow, optimal WSS) and disturbed or oscillatory flow (D-Flow, low WSS). Phase contrast images of EC in the microchannels, before and after the fluid flow exposure were captured using the Zeiss Primovert light microscope (Carl Zeiss, Bangalore, India). For visualizing the effect of varying angles of bifurcation on fluid flow patterns, magnetic beads suspended in phosphate-buffered saline were circulated through the microchannels (without cells). Live feeds of the circulating beads were recorded through Zeiss Primovert light microscope, using Apowersoft online screen recorder (https://www.apowersoft.com/free-online-screen-recorder).

**Immunofluorescence and F-actin staining**. EC were exposed to 4 h of shear stress using the coronary artery casts with varying angles of bifurcations (ranging from 30, 40, 50, 60, 70 to 80°). Following a 10 min fixation with 4% paraformaldehyde (PFA) and permeabilization using 0.1% Triton X-100 (5 min), the glass slides were blocked with 2% bovine serum albumin for 1 h; and further incubated overnight at 4 °C with eNOS Rabbit mAb (1:100, #32027), ICAM1 Rabbit pAb (1:500; #4915), CD31 (PECAM-1) Mouse mAb (1:1000; #3528) (Cell Signaling Technology, Danvers, USA), and CD144 (VE-Cadherin) Mouse mAb (1:500; #SC-9989; Santa Cruz Biotechnology, Dallas, USA) primary antibodies. The cells were then stained with Alexa Fluor™ Plus 555 and Alexa Fluor™ Plus 488 conjugated anti-Rabbit and anti-Mouse IgG secondary antibodies (1:4000; #A32732 and #A-11001), respectively, (Thermo Fisher Scientific, Mumbai, India) for 1 h and counterstained with DAPI (1 μM) for 10 min. For staining the F-actin, cells were incubated with rhodamine-tagged phalloidin (1:5000; #R415, Thermo Fisher Scientific) for 30 min, prior to DAPI staining. Fluorescence images were captured using the Zeiss Axio Scope A1 and Zeiss ApoTome.2 microscopes (Carl Zeiss). Tile scan imaging was performed for EC exposed to shear stress in 30°, 60° and 80° microchannels, and

stained for eNOS and ICAM1 using the Zeiss LSM 880 Confocal microscope (Carl Zeiss) at the Central Instrumentation Facility, BITS Pilani, Pilani Campus, India. For such imaging, a total of 375 tiles (25 horizontal × 15 vertical) along with Z-Stack (Range: 244.73 μm; Slices: 31) were scanned. Represented images were generated through the stacking of all the 31 slices using the Zeiss Zen microscopy software. The circularity of the cells and fluorescence intensities were measured using ImageJ software.

**Monocyte adhesion assay**. THP-1 cells were stained using 1,1′-Dioctadecyl-3,3,3′,3′-tetramethylindocarbocyanine perchlorate – gifted by Prof. Aniruddha Roy (Department of Pharmacy, BITS Pilani, Pilani Campus, India)—for 10 min. EC exposed to 4 h of fluid shear stress in microchannels with 30°, 60° and 80° bifurcation angles were incubated with pre-stained THP-1 monocytes ($2 \times 10^5$ cells per channel) for 30 min. Non-adherent cells were washed, and the slides were fixed using 4% PFA. DIC and fluorescence images were captured using the Zeiss LSM 880 Confocal microscope. Monocytes (red dots) adhered to the endothelial bed were counted using ImageJ software.

**Scanning electron microscopy**. Cellular morphology of EC experiencing S-Flow and D-Flow was assessed using Field Emission Scanning Electron Microscope (FE-SEM). Flow exposed HUVEC (30, 60 and 80° casts for 4 h) were fixed with 4% PFA for 10 min and dehydrated using ethanol in incremental concentrations. The slides were sputter coated with gold, and images were captured using the high vacuum mode of the FEI Apreo S (Thermo Fisher Scientific) at the Central Instrumentation Facility, BITS Pilani, Pilani Campus, India.

**Statistics and reproducibility**. All the values are expressed as the mean ± SD. Statistical differences were determined by an unpaired $t$-test for comparisons between two groups or one-way ANOVA with a Tukey post-hoc test for multiple comparisons. Statistical analyses were performed using GraphPad Prism software. A $p$-value of less than 0.05 was considered statistically significant. At least three independent biological replicates were performed for statistical analysis. In the case of immunofluorescence analysis, we have taken multiple cells from images taken for different fields from at least three independent biological replicates.

**Reporting summary**. Further information on research design is available in the Nature Research Reporting Summary linked to this article.

## Data availability
All the original immunofluorescence analysis data generated and analysed in this study are available in the supplementary section (Supplemental data 1). All other data can be obtained from the corresponding authors on reasonable request.

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

## Acknowledgements

This work was supported by the Early Career Research Award from Science and Engineering Research Board-Department of Science and Technology, Govt. of India (ECR/2017/002153) to S.M. This work was also partly supported by a Competitive Research Grant from the Department of Biotechnology, Govt. of India (BT/PR33144/MED/30/2170/2019) to S.M. We gratefully acknowledge the technical assistance of Mr. Suman Kumar (Confocal facility) and Mr. Om Prakash (FE-SEM facility).

## Author contributions

Y.T.K. designed and performed the experiments, analysed the data and wrote the first draft of the manuscript. R.B. and S.R. performed experiments and analysed data. S.K. performed computer-aided design and CFD analysis. I.N. designed and 3D-printed the models. S.M. and B.V.R.K. secured the funding, designed the experiments, supervised the study and wrote the manuscript.

## Competing interests

The authors declare no competing interests.
