## [Peer Review File · Communications Biology]

Reviewers' comments:

Reviewer #1 (Remarks to the Author):

Bifurcation angles affect local hemodynamics and have been associated with the incidence of coronary artery atherosclerosis. In this study, Katakia et al. investigated the correlation between alterations in wall shear stress (WSS), as a result of varying bifurcation angles, and changes in phenotypic markers of endothelial cells (ECs) cultured in a microfluidic device mimicking a branched human coronary artery. Their analyses showed that larger bifurcation angles are accompanied by more pronounced disturbed flow patterns (lower WSS and oscillatory flow) at branch points, leading to a rounder morphology of ECs and stronger atheroprone effect on ECs, determined by the decrease in eNOS level and increase in ICAM1 expression. Overall the findings from the in-silico analysis and in-vitro experiments are consistent with literature, and the manuscript is generally well written. However, the study does not experimentally confirm the D-Flow patterns (flow separation and oscillatory flow) nor determine the length of D-Flow regions in the microchannels with different branch angles, which therefore raises concerns about the spatial accuracy and consistency of immunofluorescence imaging in the microchannels. Other issues need to be addressed are:

1. Are the results presented in Fig. 1h-i and Supplemental Fig. 2/3 obtained from the ECs in the D-variable regions? It is not clear how the D-flow regions were selected for imaging. Were all images taken at a similar position in the microchannels with different angles (e.g., distance to the flow separation site)?
2. The conclusions of this study are heavily based upon immunofluorescence data from different regions of the microchannel. It would be more convincing if the authors could provide immunofluorescence images of the entire microchannel with the resolution to visualize the fluorescence intensity changes across regions with different flow patterns.
3. The title states "Angular difference.....governs endothelial cells structure and function.....". However, only two markers were used to assess EC states in different flow regions. No functional assays were performed to confirm the atheroprone phenotypes.
4. Flow-induced changes in gene expression and EC phenotypes in vitro are highly dependent on the duration of exposure to flow. The authors should justify why a 4-hour flow was selected for the study. Could the changes in eNOS/ICAM1 expression and EC morphology be transient responses?
5. According to Supplemental Fig. 1d, no pulse dampener was used in conjunction with the peristaltic pump. Could the liquid pulses lead to more complicated D-flow patterns in the microchannels and modulate EC phenotypes?

Reviewer #2 (Remarks to the Author):

The study is about an in-vitro and in-silico model of a coronary artery bifurcation and investigates different bifurcation angles on the wall shear stress and accumulation of plaques. While the study's goal is well-intentioned and aimed in the right direction, that is to translate computational techniques to in vitro and eventually to the clinic, there are major limitations that need to be addressed. This paper can be recommended for publication in the case authors make major changes to the manuscript.

1. It is not clear what is the major contribution of your work specially in model development and how the current study is related to the medical needs.
2. Based on the manuscript it's hard to identify the link between in silico model and in vitro study. Is there any in silico data used for the in silico model? What kind of framework that you developed?

Authors used the native ANSYS solver and it's difficult to see any contribution as development in computational modelling.

3. Despite the use of a 3D CFD simulation of a coronary artery bifurcation, the boundary conditions employed for the study are quite simple and unrealistic. The authors should have used the pulsatile flow rate at the inlet and a 3 element Windkessel model representing resistance, compliances etc. at the outlets. Then they were able to compute time average wall shear stress and oscillatory shear index which based can show a better picture of the low shear regions and the recalculation zones for governing atherogenic-endothelial dysfunction and accumulation of plaques.

4. As authors on page 9, lines 284-256 stated that "Even though the flow is laminar in the entrance region, however, it may turn into low-Re turbulent 282 in the bifurcation region. SST k- ω model³¹ was considered to achieve accurate results when fluid is in low-Re turbulence and the viscous-sublayer region." However, the low-Re turbulent models are used for the turbulent flows to capture the low-Reynolds region close to the wall where viscous effects are dominant. Based on the inlet velocity and diameter on a rough calculation the Reynolds number at the entrance is approximately 90 and by checking the maximum velocity the bifurcation with steady-state condition Reynolds number always remains very low and the choice of using the turbulent model is not correct which gives unrealistic results for the WSS calculations.

5. Despite the detailed model of in vitro, the current study adds only incremental value to the plethora of studies that are already present in the literature. For example in page 7 line 233-234 "Therefore, it would be alright to conclude that blood vessels with bifurcation angles between 50 and 60° are at a higher risk of accumulating plaques. " as a summary of your work which already can be found based on literature.

6. The explanation for in vitro setup and how this setup is different from previous studies, any novelty?

7. The choice of using non-Newtonian fluid for their in silico model is realistic. However, the medium that was used in the experiments has to be examined with a rheometer and its characteristics need to be consistent with the modified-Casson model that is used in silico.

8. The quality of the figure is very poor. Legends and titles are inconsistent. For example, in Fig 1 the contour label is blurry and cannot be read, legend unit and numbers are very inconsistent with other figures. It is highly recommended for the authors to check the journal website and see the published papers to check the quality and preparation of the figures.

9. It's highly recommended that authors add a Limitation section in the manuscript.

Response to the queries of Referee #1

Bifurcation angles affect local hemodynamics and have been associated with the incidence of coronary artery atherosclerosis. In this study, Katakia et al. investigated the correlation between alterations in wall shear stress (WSS), as a result of varying bifurcation angles, and changes in phenotypic markers of endothelial cells (ECs) cultured in a microfluidic device mimicking a branched human coronary artery. Their analyses showed that larger bifurcation angles are accompanied by more pronounced disturbed flow patterns (lower WSS and oscillatory flow) at branch points, leading to a rounder morphology of ECs and stronger atheroprone effect on ECs, determined by the decrease in eNOS level and increase in ICAM1 expression. Overall the findings from the in-silico analysis and in-vitro experiments are consistent with literature, and the manuscript is generally well written. However, the study does not experimentally confirm the D-Flow patterns (flow separation and oscillatory flow) nor determine the length of D-Flow regions in the microchannels with different branch angles, which therefore raises concerns about the spatial accuracy and consistency of immunofluorescence imaging in the microchannels. Other issues need to be addressed are:

Response: We thank the Referee for positive as well as a critical review of our manuscript. In context to the D-Flow patterns query by the referee, this information was previously shown using the *in silico* velocity vectors' data (old Figure 2a-f/ revised Figure 2a-f with higher resolution and zoomed images). However, with relatively poor representative images and without appropriate discussion of the data in the results section, the D-Flow patterns remained undiscussed in the previously submitted manuscript. In the revised version of the manuscript, we elaborated on this based on our *in silico* and experimental data (Page 6, Paragraph 2, and lines 13-23). Furthermore, we have performed new analysis and experiments to address the referees concern. We have now reported the velocity vectors using better and high resolution zoomed images (Figure 2a-f) and deliberated about the D-Flow patterns on Page 6, Paragraph 2, and lines 13-23. In addition, using real-time videography facility of an inverted microscope adapted with a video recording camera, we recorded the movement of magnetic beads through the microchannels containing 30, 60 and 80 degree bifurcation angles (Supplemental videos 1-7). Such videos clearly indicated slower movement with bead taking dips within the bifurcation region of microchannels containing 60 and 80 degree branching angles.

The Referee's points concerning the spatial determination of the D-Flow regions and the consistency of immunofluorescence imaging have been addressed further. As apparent from the representation in Figure 2o, based on the reference area as indicated by circles in WSS contour and tile scan images, the spatial accuracy and consistency have been maintained throughout the study where immunofluorescence imaging was undertaken. Moreover, we have performed tile scan imaging (that provides a broader view of the entire "Y" shaped structure) of both eNOS (Figure 3d) and ICAM1 (Figure 4d) staining of microchannels containing 30, 60 and 80 degree

branching angles, and we observed parallel findings to that of the higher magnification images. I hope these new experiments, analysis and elaborate discussion address referee's this comment.

1. Are the results presented in Fig. 1h-i and Supplemental Fig. 2/3 obtained from the ECs in the D-variable regions? It is not clear how the D-flow regions were selected for imaging. Were all images taken at a similar position in the microchannels with different angles (e.g., distance to the flow separation site)?

Response: The results presented in Figure 1n-o (old Figure 1h-i) and Supplemental Figure 2 (old Supplemental Figure 2/3) were obtained by averaging the level of eNOS and ICAM1 level from both D-Fixed and D-Variable regions (D-Flow is Average of D-Fixed and D-Variable). *In silico* WSS data (Figure 2o) was used to determine the precise location of the D-Flow regions in the microchannels. We have now included a microscopic image of the microchannel (Figure 2o) to better visualize the areas that were selected for imaging. Based on the respective WSS data, all the microscopic images were taken at similar locations in the microchannels with different angles (just after the outer-wall curvature and below the inner branch point) as depicted in Figure 2o.

To further validate the D-Flow effect and its location, we setup our flow module under an inverted microscope; circulated magnetic beads suspended in PBS, and recorded the videos at various locations (where microscopic imaging was performed throughout this study) (Supplemental Videos 1-7). As described earlier, such videos clearly indicated slower movement with bead taking dips within the bifurcation region of microchannels containing 60 and 80 degree branching angles.

2. The conclusions of this study are heavily based upon immunofluorescence data from different regions of the microchannel. It would be more convincing if the authors could provide immunofluorescence images of the entire microchannel with the resolution to visualize the fluorescence intensity changes across regions with different flow patterns.

Response: As suggested by the Referee, we captured the images of the microchannel at the lowest available magnification (5x), however, we failed to capture the entire microchannels. We therefore took 5x images of the regions precisely used for 20x imaging and fluorescence intensity analysis, and have included them to complement the respective 20x images (Supplemental Figure 4). Furthermore, to visualize a greater part of the microchannels along with the vessel branching points, we performed tile scan imaging using the confocal facility available at BITS Pilani, Pilani Campus. The tile scan images presented in Figure 3d (eNOS) and Figure 4d (ICAM1) of microchannels containing 30, 60 and 80 degree branching angles, we observed parallel findings to that observed with the higher magnification images.

3. The title states “Angular difference.....governs endothelial cells structure and function.....”. However, only two markers were used to assess EC states in different flow regions. No functional assays were performed to confirm the atheroprone phenotypes.

Response: We thank the Referee for bringing this point. The association between monocytes and atherosclerotic lesions, both in animal models and in humans, has long been recognized^{1,2}. We therefore performed monocyte adhesion assay to confirm the atheroprone phenotypes of the EC at vessel branch points. This new functional data is now included in Figure 5. This data clearly revealed highest attachment of monocyte to endothelial monolayer at D-variable regions of bifurcation in microchannel containing 60° branching points, especially in comparison to microchannel containing 30° branching points. In addition, we have further performed more structural analysis of EC by staining the cells with cell-cell contact marker CD144 (Figure 6) and by staining the cell F-actin using phalloidin-TRITC (Figure 6, Supplemental Figure 5a). Furthermore, we also performed a comprehensive structural analysis by performing additional experiments to capture SEM images of the cells to ascertain structural and surface level changes in the cells in response to differential bifurcation of the microchannels (Supplemental Figure 5a). All these data revealed significant structural deformations of the cells exposed to differential flow conditions at D-variable regions of the microchannels containing 60 and 80 degree branching angles.

4. Flow-induced changes in gene expression and EC phenotypes in vitro are highly dependent on the duration of exposure to flow. The authors should justify why a 4-hour flow was selected for the study. Could the changes in eNOS/ICAM1 expression and EC morphology be transient responses?

Response: We like to thank the referee for asking a very pertinent question. Evidence suggests that flow dependent changes in eNOS/ICAM1 expression and EC morphology are not transient. Previous findings have shown phenotypic changes in EC exposed to D-Flow for as low as 1 hour³. Similarly, our precursory publication shows how a 4 hours D-Flow exposure epigenetically regulates endothelial inflammation and apoptosis⁴. During that study we performed a time dependent D-Flow exposure (4, 8, 12, and 24 hours) experiment, using the microchannels, and assessed eNOS and ICAM1 protein levels. The changes in the expression levels were similar and comparable for all the time points. Because we observed a robust induction of endothelial inflammatory phenotype (reduction in eNOS and increase in ICAM1) within 4 hours of fluid flow exposure, we went ahead and carried out all our microchannels' experiments with a 4 hours D-Flow exposure set up.

5. According to Supplemental Fig. 1d, no pulse dampener was used in conjunction with the peristaltic pump. Could the liquid pulses lead to more complicated D-flow patterns in the microchannels and modulate EC phenotypes?

Response: We again like to appreciate the point raised by the Referee. The absence of a pulse dampener surely affects the constant flow rate. However, increasing rollers does tend to decrease the amplitude of the fluid pulsing at the outlet by increasing the frequency of the pulsed flow. The peristaltic pump that we employed had six rollers, thereby maintaining a constant-like flow rate.

Moreover, our new simulation studies with pulsatile flow shows circulatory flow patterns at the D-Flow regions, while the WSS follows a similar trend – with increasing angles of bifurcations – when compared to constant/steady-state flow (Figure 1g). Thus, our current microfluidics setup offers a more realistic *in vivo*-like fluid flow conditions.

Response to the queries of Referee #2

The study is about an in-vitro and in-silico model of a coronary artery bifurcation and investigates different bifurcation angles on the wall shear stress and accumulation of plaques. While the study's goal is well-intentioned and aimed in the right direction, that is to translate computational techniques to in vitro and eventually to the clinic, there are major limitations that need to be addressed.

This paper can be recommended for publication in the case authors make major changes to the manuscript.

We like to thank the Referee for positive review of our study and for suggesting a comprehensive plan for revising the manuscript that we believe significantly elevated to quality of the study.

1. It is not clear what is the major contribution of your work specially in model development and how the current study is related to the medical needs.

Response: Thank you for bringing this point. The *in vitro* platform employed in the study mimics *in vivo*-like fluid flow conditions. The use of 3D-printed human coronary artery for casting microchannels, proffer natural vessel curvature and disturbed *in vivo* branching point-like flow patterns. The existing eNOS and ICAM1 protein expression data along with the newly included monocyte adhesion data (Figure 5) confirms that our model is capable of inducing pro-atherogenic endothelial phenotypes. In our previous publication – employing this microchannels' model – we circulated a pharmacological inhibitor at appropriate concentrations and reported that the drug successfully inhibited EC phenotypic alterations⁴. We have further elaborated on the significance of our model and the clinical relevance of this study under the results and discussion section at Page 9, Paragraph 2, and lines 18-27.

In addition, in this paper, we study the effect of angular bifurcation on the minimum WSS using various models such as steady state, pulsatile-rigid arterial wall, and Pulsatile with Fluid-Structure-Interaction between arterial wall and fluid. We believe the model development is comprehensive, considering all the real flow scenarios. In the literature, we found studies which only consider one or the other assumption. The crux of the paper is to demonstrate and employ CFD to find the correlation between the WSS and biological response in the *in vitro* experimental set up. The framework developed includes the CAD-based geometries for both *in silico* and *in vitro* models, which would give accurate predictions of WSS and other hemodynamic descriptors.

We believe that the paper will be of value to the medical field as well since the paper (a) demonstrates a correlation between WSS and endothelial response, and (b) usage of CFD to study the flow patterns and predict the biological response.

2. Based on the manuscript it's hard to identify the link between in silico model and in vitro study. Is there any in silico data used for the in silico model? What kind of framework that you developed? Authors used the native ANSYS solver and it's difficult to see any contribution as development in computational modelling.

Response: We appreciate the comment provided by the referee. In the present study, our main aim was to use *in silico* CFD analysis with ANSYS solver and correlate such data with biological response achieved through *in vitro* cell culture based experiments to establish a correlation between the effect of angular bifurcation and the minimum WSS to that of the biological responses in cellular level. To the best of our knowledge, to date no study amalgamated these two analysis and observations together to achieve the goal of the present study; correlation of CFD based analysis with biological response in cellular level through cell culture based experimentation. CFD analysis performed in the current study allowed us to identify the regions in the bifurcation area in context to deviation in WSS with changes in branching angle and further assess the biological response based on such changes in fluid dynamics and WSS on those areas.

As for the model development, with new set of *in silico* analysis, we undertook a comprehensive effort to include all parameters, reducing the assumptions to the minimum to make it more realistic. In the revised manuscript through new sets of CFD analysis, we now report CFD analysis of steady-state simulations, pulsatile with the rigid wall, and pulsatile with the fluid-structure interaction model to achieve an exhaustive analysis to ascertain the variation in WSS and velocity streamlines between these specified models. Moreover, through such analysis, we affirm that indeed WSS at bifurcation angle remain to be proportionally altering based on the changes in the branching angles of the microchannels. In specific, WSS at the branching point of microchannel with 60° angle of bifurcation remained to be the lowest between all these newly reported analysis using different boundary conditions (Figure 1g).

3. Despite the use of a 3D CFD simulation of a coronary artery bifurcation, the boundary conditions employed for the study are quite simple and unrealistic. The authors should have used the pulsatile flow rate at the inlet and a 3 element Windkessel model representing resistance, compliances etc. at the outlets. Then they were able to compute time average wall shear stress and oscillatory shear index which based can show a better picture of the low shear regions and the recalculation zones for governing atherogenic-endothelial dysfunction and accumulation of plaques.

Response: We have now included simulation data for pulsatile flow- time average WSS (Figure 1h-m), velocity vectors (Figure 2h-m, Supplemental Figure 3a-f), and oscillatory shear index (Supplemental Figure 3g-l). Moreover, as suggested by the Referee, we performed and have

included pulsatile flow simulations' data with realistic boundary parameters (Figure 1g). All these new data have been included and discussed in the main text of the manuscript to address the referee's comment.

4. As authors on page 9, lines 284-256 stated that "Even though the flow is laminar in the entrance region, however, it may turn into low-Re turbulent 282 in the bifurcation region. SST k- ω model³¹ was considered to achieve accurate results when fluid is in low-Re turbulence and the viscous-sublayer region." However, the low-Re turbulent models are used for the turbulent flows to capture the low-Reynolds region close to the wall where viscous effects are dominant. Based on the inlet velocity and diameter on a rough calculation the Reynolds number at the entrance is approximately 90 and by checking the maximum velocity the bifurcation with steady-state condition Reynolds number always remains very low and the choice of using the turbulent model is not correct which gives unrealistic results for the WSS calculations.

Response: In the new version of the CFD studies, our inlet boundary conditions were pulsatile, which showed that there is recirculation even with a low Reynolds number. SST k-W model has been used in the literature for various studies with pulsatile flow.

5. Despite the detailed model of in vitro, the current study adds only incremental value to the plethora of studies that are already present in the literature. For example in page 7 line 233-234 "Therefore, it would be alright to conclude that blood vessels with bifurcation angles between 50 and 60° are at a higher risk of accumulating plaques. " as a summary of your work which already can be found based on literature.

Response: As pointed by the Referee, we have modified the manuscript accordingly at Page 9, Paragraph 2, last part of the discussion. The main novelty of the work is to use CFD tools to correlate the relationship between the low shear stress region in the bifurcation and find the biological response in the same region for validation. We have used CAD files for printing the *in vivo* model and the same CAD geometry is used for the CFD simulations, which enabled us to build a strong correlation study between the CFD analysis and biological response in endothelial cell culture models. To the best of our knowledge, to date no study amalgamated these two analysis and observations together to achieve the goal of the present study; correlation of CFD based analysis with biological response in cellular level through cell culture based experimentation.

6. The explanation for in vitro setup and how this setup is different from previous studies, any novelty?

Response: It has now been discussed about under the results and discussion section at Page 9, Paragraph 2, last part of the discussion. We believe the main novelty of the study lies in the model

system itself wherein we generated a computational model of a branched right human coronary artery for both 3D-printing followed by biological experimentation and CFD analysis using the same human coronary artery geometry. To date no studies, exist where they used such model system to assess the biological response based angular differences in branching point. More importantly, as stated earlier, to the best of our knowledge, to date no study amalgamated these two analysis and observations together to achieve the goal of the present study; correlation of CFD based analysis with biological response in cellular level through cell culture based experimentation.

7. The choice of using non-Newtonian fluid for their in silico model is realistic. However, the medium that was used in the experiments has to be examined with a rheometer and its characteristics need to be consistent with the modified-Casson model that is used in silico.

Response: The culture medium used for *in vitro* fluid flow exposure contained 10% fetal bovine serum; the rheological analysis of the same gave a viscosity of 2.027 mPa.s, i.e, somewhere in between that of blood (3 mPa.s) and water (1 mPa.s). Furthermore, we found the medium circulated through the microchannels to have shear-thinning properties similar to that of a Casson fluid (Figure given below).

8. The quality of the figure is very poor. Legends and titles are inconsistent. For example, in Fig 1 the contour label is blurry and cannot be read, legend unit and numbers are very inconsistent with other figures. It is highly recommended for the authors to check the journal website and see the published papers to check the quality and preparation of the figures.

Response: We thank the Referee for pointing this issue. The figures have been updated with better resolution images and newly obtained data. We hope the new figures with improved

captions and legends meet the *Communications Biology* standard. Representation of images within the figures and image quality are improved in the revised version of the manuscript.

9. It's highly recommended that authors add a Limitation section in the manuscript.

Response: As suggested by the Referee, we have added few lines stating the limitations of the study at Page 9, Paragraph 2, lines 10-18.

References

1. Gerszten, R. E. *et al.* Adhesion of Monocytes to Vascular Cell Adhesion Molecule-1–Transduced Human Endothelial Cells. *Circ. Res.* **82**, 871–878 (1998).
2. Gerhardt, T. & Ley, K. Monocyte trafficking across the vessel wall. *Cardiovasc. Res.* **107**, 321–330 (2015).
3. Higashijima, Y. *et al.* Coordinated demethylation of H3K9 and H3K27 is required for rapid inflammatory responses of endothelial cells. *EMBO J.* (2020). doi:10.15252/embj.2019103949
4. Katakia, Y. T. *et al.* Dynamic alterations of H3K4me3 and H3K27me3 at ADAM17 and Jagged-1 gene promoters cause an inflammatory switch of endothelial cells. *J. Cell. Physiol.* (2022). doi:10.1002/JCP.30579

Reviewers' comments:

Reviewer #1 (Remarks to the Author):

The updated manuscript is much improved. However, I have a few questions regarding the newly added tile scan images (i.e., Fig. 3D and 4D):

1. In 3D, it looks like the representative images for the 30 degree channel were not taken at the same focal plane as others, and the images for the 60 degree channel are probably misplaced (nuclear staining and merge).
2. In 4D, the variation of background intensities between samples makes it hard to justify the true signals. Were the images taken under the same setting? Given that the conclusion of the study is heavily based upon quantification of fluorescent images, I would suggest the authors to include some image metadata to the Materials and Methods or as part of the supplement.

Reviewer #2 (Remarks to the Author):

I am satisfied with the revised manuscript and the provided answers.

Response to the queries of Referee #1

The updated manuscript is much improved. However, I have a few questions regarding the newly added tile scan images (i.e., Fig 3D and 4D):

Response: We thank the Referee for a positive remark. We hope the following responses appropriately address the referee's concerns regarding the tile scan imaging (Figures 3D and 4D).

1. In 3D, it looks like the representative images for the 30 degree channel were not taken at the same focal plane as others, and the images for the 60 degree channel are probably misplaced (nuclear staining and merge).

Response: We thank the Referee for bringing this point. We realize that we did give out the detailed procedure of the tile scan imaging. For the tile scan image acquisition through Confocal Microscopy, we performed the tile scanning along with the Z-Stack in order to obtain the best possible stack images that represent cells which are present in larger focal planes. In so doing, we imaged a total of 375 tiles (25 horizontals and 15 vertical) with at least 31 Z-Stack slices (over a range of approximately 245 μ m) per microchannel. Because of the number stacks along with Z-scan, in many of these scans, it took us nearly 8 hours to image just one slide. We have now incorporated all the necessary information under the methodology section at Page 16, paragraph 1, and lines 15-17. In addition, for the Referee's perusal – we have provided representative 3D (Z-stack) images of the EC (exposed to fluid flow in 60 degree microchannel) stained for eNOS (Response Figure 1a) and ICAM1 (Response Figure 1b).

We request the Referee if we can only include these data as part of the response to the Referee's comments. Nonetheless, if the Referee believes that inclusion of these is essential for the current manuscript, we will include the 30 and 60 degree data in the Supplemental Figures of the

manuscript. We believe the aforementioned information (as also added to the methodology section of the revised manuscript) adequately answers the Referee's question regarding the imaging of the 30 degree microchannel. Furthermore, as pointed by the Referee, we have now rectified the image placement of the 60 degree channel in Figure 3D.

2. In 4D, the variation of the background intensities between samples makes it hard to justify the true signals. Were the images taken under the same setting? Given that the conclusion of the study is heavily based upon quantification of fluorescent images, I would suggest the authors to include some image metadata to the Materials and Methods or as part of the supplement.

Response: We appreciate the Referee's concern in pointing this out. We would like to state that all the images used for quantification were captured using the exact same laser and intensity settings. Furthermore, the variation in background intensities are only observed due to DAPI stain (blue channel) in the case of 60 degree image while background intensities almost remain constant for ICAM1 (red channel) stained images. In the revised manuscript, we have adjusted ONLY the DAPI stained image for 60 degree angle to maintain uniformity. However, in the case of tile scan imaging, the lower magnification and the huge area of interest forced us to use higher "gain" in comparison to the original 20x images used for quantification. Nonetheless, the settings were kept undisturbed for all the tile scan imaging slides. In addition, the tile scan images used in the figures 3D and 4D were captured solely with the aim of representing the entire microchannel (bifurcation region) in a single image (as suggested by the Referee in the 1st round of revision) and thus were not used for any immunofluorescence analysis. Moreover, taking the Referee's suggestion we have now included the raw data of all the immunofluorescence experiments as a supplemental excel file (previously submitted as "Master sheet" as per the journal's policy) and have mentioned about the same under the Data availability section at Page 17.

Response to the queries of Referee #2

I am satisfied with the revised manuscript and the provided answers.

Response: We gratefully thank the Referee for their critical review and valuable inputs for the overall betterment of the manuscript.